# An Analysis of the Main Nutrient Components of the Fruits of Different Macadamia (*Macadamia integrifolia*) Cultivars in Rocky Desertification Areas and a Comprehensive Evaluation of the Mineral Element Contents

Zhuanmiao Kang [1], Guangzheng Guo [1], Fengping He [1], Hui Zeng [2], Xinghao Tu [2] and Wenlin Wang [3,*]

1   Institute of Subtropical Crops, Guizhou Academy of Agricultural Sciences, Xingyi 562400, China; 18085160451@163.com (Z.K.); 18785132671@163.com (G.G.)
2   South Subtropical Crops Research Institute, Chinese Academy of Tropical Agricultural Sciences, Zhanjiang 524091, China
3   Guangxi South Subtropical Agricultural Science Research Institute, Longzhou 532415, China
*   Correspondence: 3858533@163.com; Tel.: +86-13788513949

**Abstract:** This study aims to understand the main nutrient composition and comprehensively evaluate the differences in the mineral element contents of fruits of different macadamia cultivars, as well as screen good cultivars that are suitable for use in rocky desert mountains. Nine macadamia nut cultivars were selected as test materials in rocky desert mountain orchards. The contents of crude fat, crude protein, and total soluble sugar in kernels and N, P, K, Ca, Mg, Fe, Mn, Cu, Zn, and B in peels and kernels were determined, respectively. Then, the kernels' mineral element contents were comprehensively evaluated based on principal component analysis. The results showed that the kernels were rich in crude fat, protein, and soluble sugar, with the crude fat content reaching 75% or greater, and the variation among cultivars was small. However, the variation in soluble sugar content was extensive. The content of mineral elements varied in different cultivars and parts of the fruit, with the average macronutrient content being K > N > Ca > P > Mg in the pericarp and N > K > P > Mg > Ca in the kernel, and the content of micronutrients in the pericarp and the kernel being Mn > Fe > Zn > Cu > B. By principal component analysis, the 10 mineral nutrient indexes were calculated as four principal components, with a cumulative contribution rate of 88.051%. Using the affiliation function value method and the calculation of the comprehensive evaluation value, the nine cultivars could be classified into three categories. The cultivar with the highest comprehensive evaluation value of the mineral element content was O.C. The one with the lowest value was H2, which indicated that O.C is a suitable variety for popularization in rocky desert mountainous areas. Stepwise regression analysis concluded that P, K, Fe, Mn, and Cu were the indicators significantly influencing the mineral element content of macadamia nuts and fruits in rocky desert mountains.

**Keywords:** macadamia; mineral elements; principal component analysis

## 1. Introduction

Macadamia (*Macadamia integrifolia* Maiden & Betche.), belonging to the genus Macadamia, is native to the subtropical rainforests of southeastern Queensland and northeastern New South Wales, Australia [1]. Macadamia fruit comprises the husk, shell, and kernel [2]. The kernel is rich in lipids, proteins, and important mineral nutrients [3–5]. Macadamia nut is a high-grade tree nut, and it enjoys a reputation as the "king of nuts" [6–9]. Since the introduction of macadamia nuts from Australia to China in the 1970s, due to their high economic value, stable price, and storage resistance, macadamia nuts have been rapidly spreading in China in recent years in suitable areas such as Yunnan, Guangxi, Guangdong, and Guizhou. By the end of 2020, China had become the world's largest acreage producer of macadamia nuts [10,11].

The formation of rocky desertification is a process of environmental degradation and land degradation caused by anthropogenic interference as the dominant factor in the context of the fragile environment of karst, which is the most wide-spread ecosystem in the southwestern part of China [12]. The rocky desertification covers $6.07 \times 104$ km$^2$ in the Yunnan–Guizhou–Guangxi area, accounting for 60.3% of the rocky desertification area of China, and is most prominent in the Guizhou province, the largest of the three provinces. Therefore, the planting of highly effective tree species in rocky desertification areas has been a key ecological restoration issue for local governments [13]. Macadamia nuts have gradually become a highly beneficial tree species for rocky desertification management in Guizhou, China, due to their evergreen and drought-resistant qualities and remarkable economic and ecological benefits [14]. In addition to being rich in fat and protein, macadamia nuts are also rich in various mineral elements in the fruit, which can effectively provide the necessary mineral nutrition for human growth and development. At the same time, the mineral element content is also an important index used for evaluating the quality of macadamia nuts. The current research on the mineral elements of macadamia nuts has been limited to determining their composition and content in acidic soils [15,16] and has not yet been reported in calcareous alkaline soils. Calcareous alkaline soil affects the crop's absorption and utilization of mineral nutrients due to the high pH value. At the same time, the soil has a high content of calcium, magnesium, and potassium, which is conducive to improving the quality of the crop. Compared to acidic soils, it has a low content of iron, manganese, and other trace elements, which affects the quality of the crop, and the mineral content of macadamia nuts fruits in this type of soil is unclear. With the continuous expansion of macadamia nut planting areas in rocky desertification areas, the content of mineral elements and the main nutrient status of the fruits in macadamia nuts under the unique conditions of shallow soil and infertile soil in rocky desertification areas have gradually become focuses of attention [17]. Therefore, based on the comprehensive evaluation of the differences in the fruit mineral element contents of different macadamia cultivars in rocky desertification areas, we screened macadamia cultivars with strong adaptability in rocky desertification areas, strong nutrient absorption and accumulation capacity, and excellent fruit quality. We aimed to provide a scientific basis for the management of macadamia nut fertilization in rocky desertification areas, the assessment of their mineral element content, and the comprehensive utilization of pericarp by-products, which significantly helps promote the development of the macadamia nut industry in these areas.

In this study, we used macadamia nut orchards planted in the rocky desertified mountains of Wanfenglin, Xingyi City, Guizhou Province, as the experimental objects and carried out relevant measurements and analyses on the nutritional quality of the fruits of nine macadamia nut cultivars, as well as the composition and content of the mineral elements of the pericarp and kernel, and classified and comprehensively evaluated the content of the mineral elements through the methods of principal component analysis, systematic clustering, and step-by-step regression to select excellent cultivars that have strong adaptability in rocky desertified mountains and are sufficiently rich in mineral nutrients. We also aimed to provide scientific ideas for evaluating the quality of macadamia nuts and popularizing the cultivars.

## 2. Materials and Methods

### 2.1. Experimental Materials

Fresh fruits of different cultivars that were mature and developed under normal conditions were collected from the test base of macadamia nuts planted in the Wanfenglin rocky desertification area of the Guizhou Subtropical Crops Research Institute (longitude 104°54′23″ East, latitude 24°58′30″ North, and elevation of 1250 m) (Table 1). Fruit samples were collected from 7-year-old macadamia nut trees with standard management at the base; 5 sample trees were randomly selected for each cultivar; 5 fruits with intact fruit stalks were randomly picked in the 4 directions of east, west, south, and north at the periphery of the middle- and upper part of the tree crowns; 20 ripe and fresh fruits were picked from

each sample tree, with a total of 100 ripe matured fruits for each variety; and the procedure was repeated 3 times. The samples were labeled and brought back to the laboratory for further processing.

**Table 1.** Cultivars and sources of macadamia nuts used in this study.

| No. | Cultivars | Abbreviation | Sources of Cultivars | No. | Cultivars | Abbreviations | Sources of Cultivars |
|-----|-----------|--------------|----------------------|-----|-----------|---------------|----------------------|
| 1 | Own Choice | O.C | Australia | 6 | HVA16 | A16 | Australia |
| 2 | HAES788 | 788 | America | 7 | Nanya No. 1 | NY1 | China |
| 3 | Beaumont 695 | 695 | America | 8 | Nanya No. 2 | NY2 | China |
| 4 | Hinde | H2 | Australia | 9 | Nanya No. 3 | NY3 | China |
| 5 | Guire No. 1 | GR1 | China | | | | |

The physical and chemical properties of the soil at the sample collection site were as follows: organic matter, 9.1%; effective nitrogen (alkaline-dissolved nitrogen), 249.2 mg/kg; quick-acting phosphorus, 23.53 mg/kg; quick-acting potassium, 101.68 mg/kg; exchanged calcium, 4430.2 mg/kg; exchanged magnesium, 764.46 mg/kg; and pH 7.80. The test site was at an altitude of 1250 m above sea level, and the average annual temperature was 17.5 °C, with an annual rainfall of 1000–1300 mm.

*2.2. Measurement Methods*

The macadamia nuts were cleaned, the green skin was removed, and the nuts were baked at 105 °C for 30 min. They were then dried at a constant temperature of 65 °C, crushed through a 1 mm sieve, and kept for reserve. The shelled fruits were dried at 38 °C, 48 °C, and 60 °C for 48 h. When the moisture content of the kernel was reduced to (1.5% ± 0.5%), the dried shelled fruits were broken, and the kernel was crushed through a 1 mm sieve and kept for reserve. Total soluble sugars were determined using dinitrosalicylic acid [18], crude protein was determined using Kjeldahl [19], and crude fat was determined using Soxhlet extraction [20]. The N elemental content was determined using Kjeldahl nitrogen determination [21], and P was determined using the molybdenum antimony colorimetric method; K, Ca, Mg, Fe, Mn, Cu, and Zn were determined using dry-ashing–atomic absorption; and the B elemental content was measured using the dry-ashing–curcumin colorimetric method [22] after using digestion with concentrated $H_2SO_4$-$H_2O_2$.

*2.3. Data Processing and Analysis*

The experimental data were processed and statistically analyzed using Microsoft Office (Excel) 2010 and SPSS 20.0 software, and the significance of the difference was tested using Duncan's new complex polarity method ($\alpha = 0.05$). DPS9.05 was used to perform principal component analysis (PC), analysis of the subordinate function (SF) value, cluster analysis, and stepwise regression analysis; Origin 2021 was used to plot double-labeled graphs. The related indexes were calculated [23].

The mineral nutrition of different macadamia nuts' kernels was determined for each comprehensive index affiliation's function value:

$$U(Xj) = \frac{(Xj - X\min)}{(X\max - X\min)} \quad (j = 1, 2, 3 \ldots n) \tag{1}$$

The weights of each principal component indicator were determined as follows:

$$Wj = \frac{Pj}{\sum\limits_{j=1}^{n} Pj} \quad (j = 1, 2, 3 \ldots n) \tag{2}$$

*Pj* represents the *j*th principal component. Based on the coupling of the weights of the PCs with the SF values, the composite evaluation value (*D*) was calculated with the following formula:

$$D = \sum [U(Xj) \times W(j)] \tag{3}$$

where *Xj* denotes the *j*th PC, and *X*min and *X*max denote the minimum and maximum values of the *j*th PC, respectively.

$$\text{Coefficient of variation (CV)} = \text{Standard deviation/Mean} \times 100\% \tag{4}$$

## 3. Results and Analysis

### 3.1. Analysis of the Main Nutrient Composition of Kernels of Different Macadamia Nut Cultivars

Soluble sugar, crude protein, and crude fat are important quality indicators affecting the quality of macadamia nuts. In particular, the crude fat content accounts for 70–80% of the kernel, which is the main component of macadamia nuts. In Table 2, the highest crude fat contents were in 695, H2, and NY1, while the lowest crude fat content was found in O.C However, the crude fat content of all the cultivars was more than 72%, which is in line with the basic quality requirements of macadamia nuts, and the coefficient of variation among cultivars was small, being only 1.90%. The contents of soluble sugar and protein are closely related to the nutrition and taste of macadamia nuts. As shown in Table 2, the soluble sugar content varied greatly among different cultivars, with the coefficient of variation reaching 13.32%. In contrast, the coefficient of variation for crude protein was relatively small, only 4.72%. The soluble sugar content of 788 reached 45.36 mg/kg, 23.83% higher than the average, which is 1.39 times higher than that of the lowest content of A16 (32.64 mg/kg), and this may be the reason why 788 has a light and sweet flavor. The highest crude protein content was found in NY2, at 8.42%, which was significantly higher than the other cultivars. The lowest content was found in A16, at 7.38%. The crude protein content of NY2 was 1.14 times higher than that of A16.

**Table 2.** Differential analysis of the content of major nutrients in kernels of different macadamia nut cultivars.

| Cultivars | Crude Fat (%) | Crude Protein (%) | Total Soluble Sugar (mg/kg) |
|---|---|---|---|
| O.C | 75.55 ± 0.70 c | 7.91 ± 0.10 b | 34.96 ± 2.16 b |
| 788 | 76.94 ± 1.00 b | 7.41 ± 0.06 c | 45.36 ± 5.12 a |
| 695 | 79.38 ± 0.53 a | 7.71 ± 0.34 bc | 36.77 ± 3.11 b |
| H2 | 79.30 ± 0.21 a | 7.88 ± 0.06 b | 35.7 ± 2.11 b |
| GR1 | 76.34 ± 0.66 bc | 7.71 ± 0.13 bc | 38.53 ± 4.08 b |
| A16 | 77.05 ± 0.44 b | 7.38 ± 0.04 c | 32.64 ± 3.27 b |
| NY1 | 78.99 ± 0.53 a | 8.07 ± 0.20 ab | 36.29 ± 3.47 b |
| NY2 | 76.96 ± 0.33 b | 8.40 ± 0.42 a | 36.33 ± 3.02 b |
| NY3 | 76.22 ± 0.72 bc | 7.97 ± 0.41 b | 32.92 ± 0.93 b |
| Mean value | 77.41 ± 1.47 | 7.83 ± 0.37 | 36.61 ± 4.51 |
| CV/% | 1.90 | 4.72 | 12.32 |

Note: Data are expressed as mean ± standard deviation, and different lowercase letters after the data in the same column indicate significant differences between cultivars ($p < 0.05$).

### 3.2. Mineral Element Characterization of Different Macadamia Nut Pericarps in Rocky Desertification Areas

As shown in Table 3, the average content of mineral elements in the pericarp of the nine cultivars, in descending order, was K > N > Ca > P > Mg for macroelements and Mn > Fe > Zn > Cu > B for microelements. There were differences in the content of mineral elements in the pericarp of different cultivars, with the highest N content of 788 being 7.42 g/kg, which was significantly higher than the other cultivars. Additionally, the highest P content in macroelements was found in 788 at 1.17 g/kg, and the lowest was found in NY1. The highest K content was found in GR1 at 24.54 g/kg, and the lowest was found in NY1, only 11.09 g/kg. The highest Ca content was found in A16, at 2.21 g/kg, and

the lowest was found in NY1 and NY2, both at 0.87 g/kg; the highest Mg content was 1.02 g/kg in NY3, and the lowest was 0.35 g/kg in 788. Moreover, regarding trace elements, the highest Fe content was 71.74 mg/kg in 695, and the lowest was 34.16 mg/kg in NY1. The highest Mn content was 268.22 mg/kg in NY3, and the lowest was 88.06 mg/kg in NY2; the highest Cu content was in 695, and the lowest was in NY2; the highest Zn content was in NY1, and the lowest was in NY2. The highest B content was in O.C., and the lowest was in A16.

**Table 3.** Differences in mineral nutrient content in the pericarp of different cultivars of macadamia nuts.

| Cultivars | N (g/kg) | P (g/kg) | K (g/kg) | Ca (g/kg) | Mg (g/kg) |
|---|---|---|---|---|---|
| O.C | 4.73 ± 0.60 b | 0.84 ± 0.05 b | 20.19 ± 0.79 bc | 1.44 ± 0.10 b | 0.64 ± 0.01 c |
| 788 | 7.42 ± 0.40 a | 1.17 ± 0.06 a | 15.91 ± 1.03 d | 1.05 ± 0.32 c | 0.35 ± 0.01 e |
| 695 | 4.59 ± 0.60 b | 0.84 ± 0.04 b | 16.81 ± 1.74 d | 1.64 ± 0.11 b | 0.53 ± 0.04 d |
| H2 | 4.95 ± 0.28 b | 0.64 ± 0.03 c | 17.18 ± 0.49 cd | 2.04 ± 0.21 a | 0.54 ± 0.02 d |
| GR1 | 4.98 ± 0.51 b | 0.86 ± 0.16 b | 24.54 ± 3.28 a | 0.82 ± 0.13 c | 0.66 ± 0.04 c |
| A16 | 4.67 ± 1.04 b | 0.67 ± 0.04 c | 17.16 ± 1.68 cd | 2.21 ± 0.21 a | 0.80 ± 0.08 b |
| NY1 | 5.20 ± 0.59 b | 0.54 ± 0.01 c | 11.09 ± 0.56 e | 0.87 ± 0.01 c | 0.55 ± 0.05 d |
| NY2 | 4.63 ± 0.36 b | 0.56 ± 0.06 c | 15.44 ± 2.95 d | 0.87 ± 0.01 c | 0.55 ± 0.05 d |
| NY3 | 5.44 ± 1.00 b | 1.13 ± 0.08 a | 22.79 ± 0.74 ab | 1.01 ± 0.25 c | 1.02 ± 0.07 a |
| Mean value | 5.18 ± 0.89 | 0.81 ± 0.23 | 17.90 ± 4.06 | 1.33 ± 0.53 | 0.63 ± 0.19 |
| CV/% | 17.12 | 28.48 | 22.69 | 39.85 | 30.60 |
| Cultivars | Fe (mg/kg) | Mn (mg/kg) | Cu (mg/kg) | Zn (mg/kg) | B (mg/kg) |
| O.C | 57.87 ± 8.32 abc | 208.03 ± 4.95 b | 6.94 ± 0.95 b | 9.45 ± 3.05 cd | 11.70 ± 2.57 a |
| 788 | 54.83 ± 5.79 bcd | 88.30 ± 22.94 e | 7.87 ± 0.45 b | 11.28 ± 3.76 bcd | 9.68 ± 2.07 abc |
| 695 | 71.74 ± 14.14 a | 276.65 ± 2.00 a | 9.57 ± 0.23 a | 13.29 ± 1.35 bc | 8.53 ± 2.29 bcd |
| H2 | 51.03 ± 4.53 bcd | 141.78 ± 8.49 c | 7.08 ± 0.49 b | 17.84 ± 1.58 a | 7.32 ± 0.97 cd |
| GR1 | 71.36 ± 10.01 a | 92.71 ± 6.25 e | 6.49 ± 0.55 bc | 10.73 ± 2.10 bcd | 11.16 ± 0.41 ab |
| A16 | 40.64 ± 3.92 de | 155.84 ± 5.36 c | 6.61 ± 1.33 bc | 14.10 ± 1.42 b | 5.84 ± 0.43 d |
| NY1 | 34.16 ± 4.88 e | 114.18 ± 4.21 d | 7.04 ± 0.25 b | 18.55 ± 2.41 a | 6.08 ± 1.27 d |
| NY2 | 47.77 ± 3.84 cde | 88.06 ± 8.52 e | 5.12 ± 1.69 c | 7.69 ± 0.68 d | 7.34 ± 0.34 cd |
| NY3 | 64.54 ± 8.06 ab | 268.22 ± 11.00 a | 6.96 ± 0.65 b | 8.95 ± 1.28 d | 7.10 ± 0.85 cd |
| Mean value | 54.88 ± 13.02 | 159.31 ± 74.93 | 7.07 ± 1.19 | 12.43 ± 3.84 | 8.30 ± 2.12 |
| CV/% | 23.72 | 47.03 | 16.75 | 30.88 | 25.54 |

Note: Data are expressed as mean ± standard deviation, and different lowercase letters after the data in the same column indicate significant differences between cultivars ($p < 0.05$).

Table 3 shows that the nutrient components of macadamia nut pericarp varied. The most significant variation in macro- and mesoelements was Ca, with a coefficient of variation of 39.85%, and the smallest was N, with a coefficient of variation of 17.12%. The most significant variation in trace elements was Fe, with a coefficient of variation of 47.03%, and the smallest was Cu, with a coefficient of variation of 16.75%.

*3.3. Mineral Nutritional Characterization of Different Macadamia Nut Kernels in Rocky Desertification Areas*

As shown in Table 4, the average content of mineral elements in the kernels of the nine cultivars, in descending order, was N > K > P > Mg > Ca for macroelements and Mn > Fe > Zn > Cu > B for microelements. The content of mineral elements in the kernels varied among cultivars, with the highest N content being found in A16, followed by O.C., with 14.95 g/kg and 14.75 g/kg, respectively. The lowest content was found in H2 and was 11.55 g/kg. Additionally, regarding macro- and mesoelements, the highest P content was found in O.C, at 2.98 g/kg, and the lowest was found in 695, at 1.19 g/kg; the highest K content was found in NY3, at 4.55 g/kg, and the lowest was found in 695, at 2.92 g/kg. The highest Ca content was found in O.C, at 0.98 g/kg, and the lowest was found in H2, at 0.98 g/kg; the highest Mg content was found in NY1, at 1.76 g/kg, while the lowest content was found in 695, at 0.98 g/kg. Furthermore, regarding trace elements, O.C had the highest Fe content, while H2 had the lowest Fe content; the highest Mn content was found in GR1, at 101.85 mg/kg, while the lowest content was found in 788, at 40.81 mg/kg. The highest Cu content was found in NY1, while the lowest was found in H2; the highest Zn content was found in A16, while the lowest was found in NY3; the highest content of

B was found in 695, and the lowest was found in 788. There were different variations in different mineral elements in the kernels, among which the smallest coefficient of variation in macro- and mesoelements was N, 9.47%. The highest coefficient of variation was P, with 22.05%. In comparison, the lowest coefficient of variation in trace elements was Cu, with 12.33%, and the highest was B, with 77.97%. Thus, the variation in trace elements was more significant among cultivars.

**Table 4.** Differences in mineral nutrient content of kernels in different macadamia cultivars.

| Cultivars | N (g/kg) | P (g/kg) | K (g/kg) | Ca (g/kg) | Mg (g/kg) |
|---|---|---|---|---|---|
| O.C | 14.75 ± 1.02 a | 2.98 ± 0.52 a | 4.05 ± 0.46 ab | 0.98 ± 0.06 a | 1.16 ± 0.06 ab |
| 788 | 13.75 ± 1.49 ab | 2.29 ± 0.09 ab | 3.85 ± 0.49 abcd | 0.93 ± 0.10 ab | 1.01 ± 0.11 ab |
| 695 | 11.68 ± 1.23 ab | 1.19 ± 0.44 c | 2.92 ± 0.24 d | 0.55 ± 0.09 de | 0.98 ± 0.11 b |
| H2 | 11.55 ± 1.12 b | 2.40 ± 0.05 ab | 3.00 ± 0.17 cd | 0.35 ± 0.06 e | 1.12 ± 0.07 ab |
| GR1 | 14.07 ± 0.83 ab | 2.23 ± 0.26 ab | 4.00 ± 0.34 abc | 0.67 ± 0.07 cd | 1.30 ± 0.10 ab |
| A16 | 14.95 ± 2.11 a | 2.21 ± 0.10 ab | 3.50 ± 0.08 bcd | 0.51 ± 0.15 de | 1.25 ± 0.10 ab |
| NY1 | 12.58 ± 1.14 ab | 1.91 ± 0.24 bc | 3.81 ± 0.43 abcd | 0.79 ± 0.11 bc | 1.76 ± 0.98 a |
| NY2 | 14.33 ± 2.62 ab | 2.61 ± 0.97 ab | 3.00 ± 0.46 cd | 0.53 ± 0.16 de | 1.22 ± 0.09 ab |
| NY3 | 12.96 ± 0.02 ab | 2.21 ± 0.54 ab | 4.55 ± 1.19 a | 0.60 ± 0.09 cd | 1.46 ± 0.15 ab |
| Mean value | 13.40 ± 1.27 | 2.23 ± 0.49 | 3.63 ± 0.56 | 0.66 ± 0.21 | 1.25 ± 0.24 |
| CV/% | 9.47 | 22.05 | 15.53 | 31.41 | 19.19 |

| Cultivars | Fe (mg/kg) | Mn (mg/kg) | Cu (mg/kg) | Zn (mg/kg) | B (mg/kg) |
|---|---|---|---|---|---|
| O.C | 36.51 ± 5.09 a | 66.72 ± 22.26 b | 7.23 ± 0.25 ab | 14.67 ± 1.54 bcd | 2.50 ± 0.36 c |
| 788 | 36.24 ± 2.00 a | 40.81 ± 1.97 b | 7.52 ± 1.29 ab | 13.32 ± 4.05 cd | 1.82 ± 0.46 c |
| 695 | 30.66 ± 5.88 ab | 99.87 ± 24.40 a | 6.06 ± 1.56 b | 13.31 ± 0.97 cd | 15.85 ± 0.23 a |
| H2 | 21.22 ± 5.00 c | 41.30 ± 8.52 b | 5.58 ± 1.58 b | 18.50 ± 2.84 b | 9.27 ± 4.46 b |
| GR1 | 30.79 ± 2.55 ab | 101.85 ± 25.93 a | 7.71 ± 0.39 ab | 19.61 ± 1.51 b | 4.22 ± 1.96 c |
| A16 | 33.66 ± 7.60 ab | 54.75 ± 4.98 b | 6.72 ± 1.05 ab | 27.91 ± 1.39 a | 3.69 ± 0.93 c |
| NY1 | 26.35 ± 2.02 bc | 41.00 ± 5.79 b | 8.32 ± 0.52 a | 18.16 ± 1.71 bc | 4.55 ± 1.71 c |
| NY2 | 31.64 ± 3.61 ab | 48.52 ± 4.49 b | 6.51 ± 1.14 ab | 14.64 ± 5.34 bcd | 5.02 ± 0.87 c |
| NY3 | 26.28 ± 1.71 bc | 58.14 ± 10.02 b | 6.71 ± 1.40 ab | 11.71 ± 1.55 d | 3.60 ± 0.60 c |
| Mean value | 30.37 ± 5.02 | 61.44 ± 24.00 | 6.93 ± 0.85 | 16.87 ± 4.94 | 5.61 ± 4.38 |
| CV/% | 16.53 | 39.06 | 12.33 | 29.30 | 77.97 |

Note: Data are expressed as mean ± standard deviation, and different lowercase letters after the data in the same column indicate significant differences between cultivars ($p < 0.05$).

*3.4. Correlation Analysis of Mineral Element Contents of Different Macadamia Nuts in Rocky Desertification Areas*

As seen in Figure 1, there were significant correlations between the contents of different mineral elements in the pericarp and kernel. N in the pericarp was significantly positively correlated with P in the pericarp and Ca in the kernel. P in the pericarp was significantly positively correlated with K, Fe, and B in the pericarp and with K and Ca in the kernel; and it was significantly negatively correlated with Zn in the pericarp and kernel. Moreover, K in the pericarp was significantly positively correlated with Mg in the pericarp and Mn in the kernel, significantly positively correlated with Fe in the pericarp, and significantly negatively correlated with Zn in the pericarp. Ca in the pericarp was significantly positively correlated with Mn in the pericarp and Cu in the kernel. In contrast, Mg in the pericarp was significantly positively correlated with B in the pericarp and K in the kernel. Furthermore, Fe in the pericarp was significantly positively correlated with Mn and Cu in the pericarp and positively correlated with B in the pericarp and Mn in the kernel. Mn in the pericarp was significantly positively correlated with Cu in the pericarp and B in the kernel. In contrast, Cu in the pericarp was positively correlated with B in the kernel. Zn in the pericarp was significantly positively correlated with B in the pericarp, N in the kernel, and Fe, while B in the pericarp was significantly positively correlated with Ca and Mn in the kernel.

N in the kernel was significantly positively correlated with P and K in the kernel, and N, P, and K in the kernel were significantly negatively correlated with B. Ca in the kernel was significantly negatively correlated with Fe in the kernel. In addition, Fe in the kernel was significantly negatively correlated with Zn in the peel, Mn in the kernel

was significantly positively correlated with B in the kernel, and Cu in the kernel was significantly negatively correlated with B in the kernel.

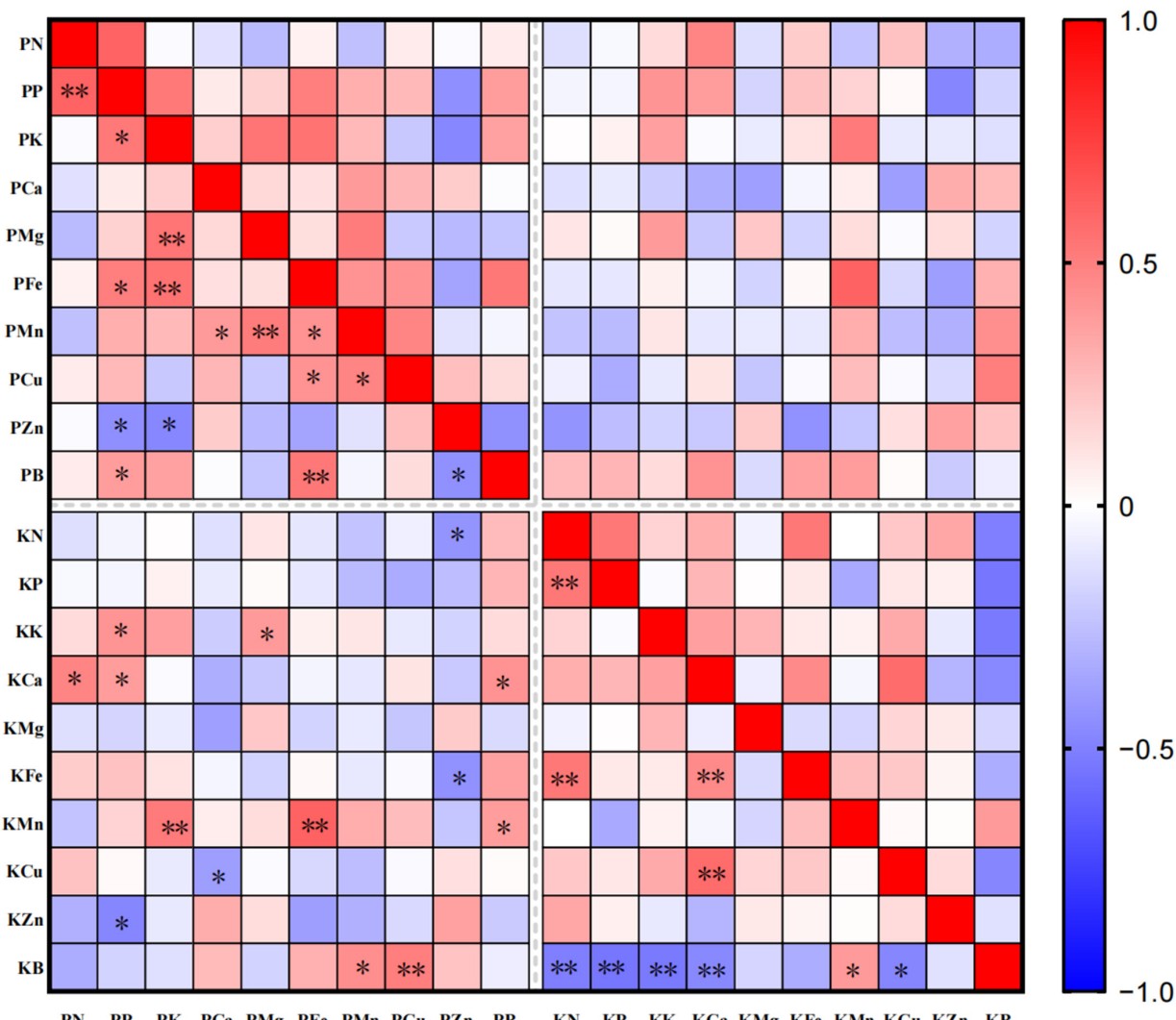

**Figure 1.** Correlation analysis of each mineral element between the peel (referred to as P) and kernel (referred to as K); ** represents a highly significant correlation ($p < 0.01$), and * represents a significant correlation ($p < 0.05$).

*3.5. Principal Component Analysis of Mineral Element Contents of Different Macadamia Nut Kernels in Rocky Desertification Areas*

The contents of 10 mineral elements in nine macadamia nut kernels were subjected to principal component analysis. As shown in Table 5, according to the principle that the cumulative contribution rate is greater than 80% [24], the cumulative contribution rate of the first four principal components was 88.051%. Therefore, by summarizing the contents of these mineral elements using the first four principal components, a comprehensive evaluation of the minerals' nutritional status in the nine macadamia nut kernels could be made. Using PCA analysis, the 10 mineral element indicators were divided into four independent major components, PC1, PC2, PC3, and PC4, and their contribution rates were 40.856%, 18.803%, 16.352%, and 12.040%, respectively. Their eigenroots were 4.086, 1.880, 1.635, and 1.204, respectively. In the loadings of PC1 values, it can be seen that N, P, K, Ca, and Cu have higher positive loadings, and B has higher negative loadings; principal component PC2 shows that Fe and Mn have higher positive loadings, while Mg has higher negative values. In addition, in principal component PC3, Ca, Mn, and Cu all have higher

positive loadings, and P and Zn have higher negative loadings. In contrast, in principal component PC4, Cu and Zn have high positive loading values, and P has high negative loading values.

**Table 5.** Eigenvalues and contribution rates of each comprehensive index.

| Principle Factor | | PC1 | PC2 | PC3 | PC4 |
|---|---|---|---|---|---|
| Eigenvalues | | 4.086 | 1.880 | 1.635 | 1.204 |
| Contribution ratio/% | | 40.856 | 18.803 | 16.352 | 12.040 |
| Cumulative contribution ratio/% | | 40.856 | 59.659 | 76.011 | 88.051 |
| | N | **0.384** | 0.285 | −0.274 | 0.229 |
| | P | **0.306** | 0.059 | **−0.434** | **−0.346** |
| | K | **0.360** | −0.207 | 0.274 | −0.042 |
| | Ca | **0.384** | 0.143 | **0.361** | −0.164 |
| Eigenvector | Mg | 0.165 | **−0.623** | 0.096 | 0.231 |
| | Fe | 0.291 | **0.549** | 0.062 | 0.104 |
| | Mn | −0.115 | **0.331** | **0.416** | **0.430** |
| | Cu | **0.380** | −0.187 | **0.313** | 0.248 |
| | Zn | 0.025 | −0.069 | **−0.456** | **0.695** |
| | B | **−0.460** | 0.118 | 0.201 | 0.074 |

Note: for the data shown in bold font, the absolute values of the eigenvectors are greater than 0.300 for each comprehensive index.

In addition, the double-scaled plots of PC1 and PC2 in Figure 2 show that some of the indicators have similar positions, which indicates a strong correlation or some overlap of information between the indicators' information. We can also see that Zn makes a relatively small contribution to the principal components. In contrast, the major and intermediate elements, such as N, P, K, and Ca, are highly distinguishable and representative indicators, indicating that they play a vital role. In the distribution points of different cultivars, all the samples are more concentrated, indicating that their mineral element contents are similar, and the repeatability of the sampling is better.

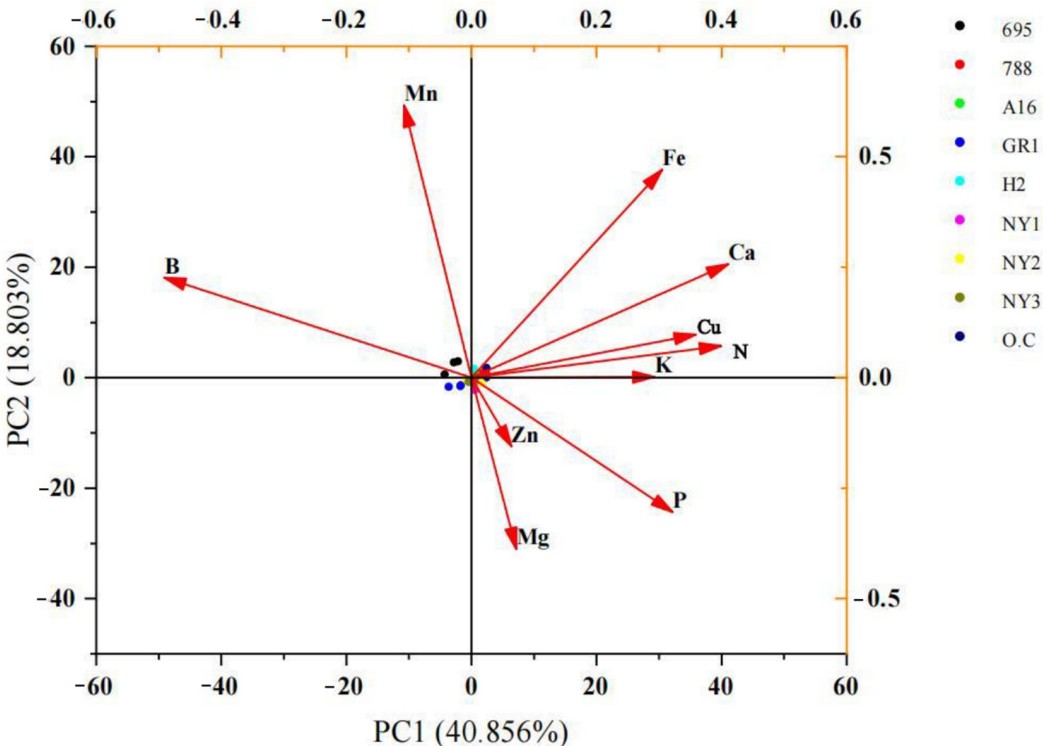

**Figure 2.** AMMI model (PC1 and PC2) and bi-labeled plots.

*3.6. Comprehensive Evaluation of Mineral Element Contents of Different Macadamia Nuts' Fruits*

3.6.1. Analysis of Affiliation Function Values

According to the factor scores obtained from the principal component analysis (Table 6), in PC1, O.C had the highest affiliation value of 1.000, while 695 had the lowest value of 0.000; in PC2, 695 had the highest value, and NY1 had the lowest; in PC3, 695 had the highest value, and A16 had the lowest; and in PC4, A16 had the highest value, and 788 had the lowest.

**Table 6.** Composite indexes, index weights, U (Xj), and composite evaluation values of nine macadamia cultivars' mineral D contents.

| Cultivar | PC1 | PC2 | PC3 | PC4 | U (X1) | U (X2) | U (X3) | U (X4) | D | Rank |
|---|---|---|---|---|---|---|---|---|---|---|
| O.C | 2.612 | 1.404 | 0.116 | −0.762 | 1.000 | 0.968 | 0.543 | 0.127 | 0.789 | 1 |
| 788 | 1.813 | 1.054 | 0.464 | −1.153 | 0.877 | 0.886 | 0.632 | 0.000 | 0.713 | 3 |
| 695 | −3.872 | 1.543 | 1.907 | 0.439 | 0.000 | 1.000 | 1.000 | 0.516 | 0.470 | 8 |
| H2 | −3.099 | −1.001 | −1.681 | −1.001 | 0.119 | 0.406 | 0.085 | 0.049 | 0.165 | 9 |
| GR1 | 0.887 | 0.270 | 0.794 | 1.535 | 0.734 | 0.703 | 0.716 | 0.871 | 0.743 | 2 |
| A16 | 0.520 | 0.426 | −2.013 | 1.932 | 0.677 | 0.739 | 0.000 | 1.000 | 0.609 | 4 |
| NY1 | 0.920 | −2.742 | 0.972 | 0.596 | 0.739 | 0.000 | 0.761 | 0.567 | 0.562 | 5 |
| NY2 | −0.147 | 0.579 | −1.338 | −0.676 | 0.575 | 0.775 | 0.172 | 0.154 | 0.485 | 7 |
| NY3 | 0.367 | −1.532 | 0.779 | −0.911 | 0.654 | 0.282 | 0.712 | 0.078 | 0.507 | 6 |
| Weight | 0.409 | 0.188 | 0.164 | 0.120 | 0.464 | 0.214 | 0.186 | 0.137 | | |

3.6.2. Calculation of Indicator Weights and Comprehensive Evaluation Value of Mineral D Element Content

Tables 5 and 6 show that the contribution rates of PC1, PC2, PC3, and PC4 were 40.856%, 18.803%, 16.352%, and 12.040%, respectively, and the weights of the indexes were 0.464, 0.214, 0.186, and 0.137, respectively. The D value is an index reflecting a high or low content of mineral elements in the kernels of macadamia nuts. The higher the D value is, the higher the content of mineral elements in the kernel is. Table 5 shows that O.C has the highest D value, followed by GR1, 788, A16, NY1, NY3, NY2, and 695, while H2 has the lowest D value. These results indicate that O.C kernels had the most adequate content of mineral elements, while H2 had a relatively low content.

3.6.3. Cluster Analysis

Firstly, we clustered the kernels of each cultivar according to their mineral element contents (Figure 3A). Based on the sum-of-squares method of deviation, the mineral element contents of the kernels of nine cultivars could be divided into three categories: the first category included O.C, 788, A16, and NY2; the second category included GR1, NY1, and NY3; and the third category included H2 and 695. Secondly, the nine cultivars of macadamia kernel mineral elements were clustered and analyzed according to the composite evaluation value D (Figure 3B), and using the longest distance method, they could be classified into three categories, with O.C, GR1, and 788 in the first category; A16, NY1, NY3, NY2, and 695 in the second category; and H2 in the third category.

3.6.4. Stepwise Regression Analysis

To establish the evaluation model of the mineral element content in macadamia nut kernels—and thus further understand the relationship between the mineral element content and each index—and screen the accurate evaluation indexes of the mineral element abundance and deficiency in macadamia nut kernels, according to the principle of the maximum coefficient of determination, $R^2$ and $p < 0.05$, we can obtain the regression equation, $D = -1.1794 + 0.0351P + 0.0738K + 0.0221Fe + 0.0019Mn + 0.0877Cu$, $R^2 = 0.997$, $p = 0.0003$. According to this equation, P and K are the indexes of macroelements that play a vital role in the mineral nutrition of macadamia nut kernels. At the same time, Fe, Mn, and Cu are very important indexes of trace elements.

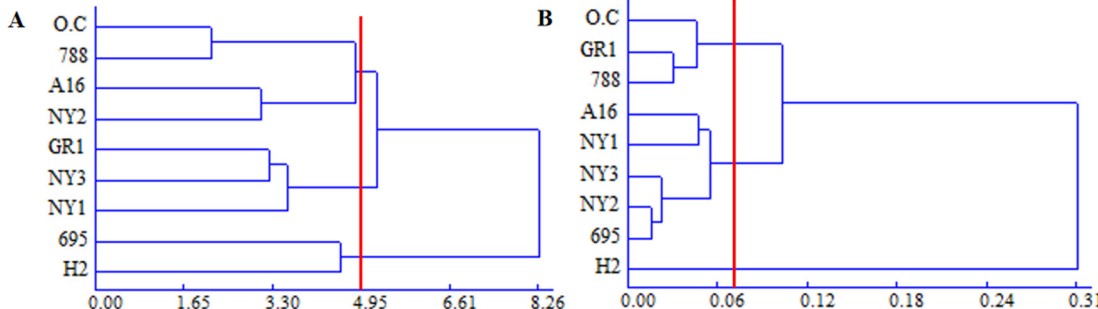

**Figure 3.** Cluster analysis. (**A**) Clustering is based on the content of each mineral element; (**B**) clustering is based on the D value of the composite rating.

3.6.5. Characterization of Kernel Mineral Content in Different Classes of Macadamia Cultivars

Combining the results of the cluster analysis and stepwise regression analysis to analyze the results of the mineral element content of kernels from each cultivar, we found that the average content of P, K, Fe, Mn, and Cu did not differ significantly among groups, indicating that these indexes had a high degree of inter-varietal variability within each group. Group 1 had a high content of P and Fe, while group 3 had a high content of K and Mn (Table 7).

**Table 7.** Description of each cluster in the hierarchical cluster result.

| Cluster | P (g/kg) | K (g/kg) | Fe (mg/kg) | Mn (mg/kg) | Cu (mg/kg) |
|---|---|---|---|---|---|
| 1 | 2.50 ± 0.42 a | 3.97 ± 0.10 a | 34.52 ± 3.22 a | 69.79 ± 30.63 a | 7.49 ± 0.24 a |
| 2 | 2.03 ± 0.53 a | 3.56 ± 0.66 a | 29.72 ± 3.29 a | 60.46 ± 22.98 a | 6.86 ± 0.86 a |
| 3 | 2.23 ± 0.26 a | 4.00 ± 0.34 a | 30.79 ± 2.55 a | 101.85 ± 25.93 a | 7.71 ± 0.39 a |

Note: Data are expressed as mean ± standard deviation, and different lowercase letters after the data in the same column indicate significant differences between cultivars ($p < 0.05$).

**4. Discussion**

*4.1. Major Nutrient Contents and Differences in Macadamia Kernels*

The macadamia nut kernel is the edible part of the fruit, which contains more than 70% crude fat. There may be differences in the crude fat content of macadamia nuts in different regions; for example, Kaijser et al. studied the chemical composition of four macadamia varieties and found that macadamia kernels contained 69.1–78.4% lipids [25]. Castilho Maro et al. studied the chemical composition of 22 Brazilian macadamia cultivars, and the lipid content of macadamia kernels ranged from 33.13 to 64.28% [26]. According to the macadamia breeding program in China, the kernel must have a crude fat content of 72% or more [27]. In this study, we found that all the macadamia nut cultivars with a crude fat content above 75% reached the expected target, which meets the basic requirements for the quality of commercial macadamia nut fruits, indicating that this region is suitable for macadamia nut cultivation. At the same time, we also paid attention to the contents of soluble and crude protein, which are closely related to the taste and nutrition of macadamia nuts, and compared with the crude fat content, the variability between its varieties is larger. This also lays the foundation for our subsequent selection of cultivars with specific tastes. In conclusion, from the point of view of the main nutritional components of macadamia nuts, it is feasible to plant macadamia nuts in rocky desertification areas.

*4.2. Mineral Element Content and Differences in Macadamia Nut Fruits*

Mineral elements are the basis of fruits' growth, development, yield formation, and quality improvement. Macadamia nut pericarp is a by-product that is left after the primary processing of macadamia nuts, which is almost useless [28]. Therefore, whether there are major or trace elements, they play a significant role in the growth and development of

the pericarp. The results of this study showed that the average contents of five macronutrients and five trace elements in the pericarp of the test cultivars were K > N > Ca > P > Mg and Mn > Fe > Zn > Cu > B in descending order, among which the K content was 13.46–28.58 times higher than the contents of Ca, P, and Mg, and the contents of Fe and Mn were 4.41–22.52 times higher than the contents of Cu and Zn. These are the more dominant macromineral and micromineral elements in the pericarp. An analysis of the mineral content of the pericarp of other nut species such as walnut also showed that the content of K in the pericarp at the maturity stage was significantly higher than other mineral elements [29], which was consistent with the results of the present study, and also indicated that K might be the most abundant mineral element accumulated in the pericarp of the woody nut. Regarding China's planting of macadamia nuts, after the bumper crop, the production of peels can reach more than 0.3 million tons due to macadamia nut processing plants, and how to deal with these peels is a problem worthy of study. This study found that the peel contains rich mineral elements, and many elements are present in quantities exceeding 3%; further, it also contains a wealth of trace elements. Therefore, using the peel as a raw material for organic fertilizer is a good option.

Macadamia nut kernels are rich in mineral nutrient composition, with balanced ratios, which can meet the human body's needs for various essential trace elements, and are an excellent food source for the human body to supplement trace elements. The average contents of macronutrients and trace elements, in descending order, are N > K > P > Mg > Ca and Mn > Fe > Zn > Cu > B. This result generally agrees with the findings of the mineral nutrition of South African macadamia seed kernels [30]. The content of mineral elements in the kernels in this study was also similar to the findings of Tan et al. in acidic soils, suggesting that the uptake of mineral elements in macadamia nuts is normal when grown in calcareous soils [31]. It is also similar to the mineral content of other nut seed kernels [32]. Some studies have shown that the average intake of K for adults is about 2250 mg·d$^{-1}$, that of Ca is 800 mg·d$^{-1}$, that of Mg is 325 mg·d$^{-1}$, that of Zn is 12.5 mg·d$^{-1}$, that of Mn is 6 mg·d$^{-1}$, that of Mn is 6.5 mg·d$^{-1}$, that of Ca is 800 mg·d$^{-1}$, that of Mg is 325 mg·d$^{-1}$, that of Zn is 12.5 mg·d$^{-1}$, that of Mn is 6 mg·d$^{-1}$, that of Fe is 14 mg·d$^{-1}$, and that of Cu is 2 mg·d$^{-1}$ [33]. Based on the average of the results of this study, an adult consuming five to six macadamia nuts (approximately 15 g) per day would meet 2.42% of the daily intake of K, 1.23% of Ca, 5.77% of Mg, 2.02% of Zn, 15.36% of Mn, 3.25% of Fe, and 5.20% of Cu.

The absorption and accumulation status of mineral elements in the pericarp and kernel are affected by the interaction between various mineral elements in the same site and the absorption of mineral elements in different sites. The correlation analysis of each mineral element in the pericarp and kernel can provide a theoretical basis for utilizing mineral elements in macadamia nuts. This study showed that 10 mineral elements in the pericarp had significant or highly significant correlations with some mineral elements in the pericarp and kernel. For example, P in the pericarp was significantly positively correlated with K, Fe, and B in the pericarp and K and Ca in the kernel and significantly negatively correlated with Zn in the pericarp and kernel, which indicated that the absorption of pericarp mineral nutrients affected the kernel of the fruit and thus played an important role. At the same time, the mineral elements in the kernel were affected by the mineral nutrition of the pericarp and the mineral elements inside the kernel. For example, N in the kernel was significantly positively correlated with P and Fe, and N, P, and K in the kernel were significantly negatively correlated with B.

### 4.3. Comprehensive Evaluation of Mineral Element Contents of Macadamia Nut Kernels

Principal component analysis (PCA), which utilizes the idea of dimensionality reduction to transform multiple inter-related indicators into a small number of correlated or independent indicators [34,35], has been widely used in the study of genetic diversity of germplasm resources of horticultural crops, such as mustard, Avena sativa, olives, walnuts, and pears [31]. This study used principal component analysis to comprehensively evaluate 10 mineral elements, N, P, K, Ca, Mg, Fe, Mn, Cu, Zn, and B, in nine macadamia nut kernels

grown in rocky, deserted mountainous areas. Four new composite indexes were used to reflect 88.051% of the information contained in the original indexes. An analysis of the affiliation function value was used to determine the comprehensive evaluation value of the mineral element content D, which objectively responded to the mineral element content level of the kernels of nine macadamia nut cultivars. In the cluster analysis, the mineral element content levels of the nine cultivars were divided into three categories. The results showed that the kernels of O.C, 788, and GR1 had the highest comprehensive mineral element content, indicating that their ability to absorb and accumulate mineral elements in the fruit in the rocky desertification area was relatively strong, while the comprehensive evaluation value of H2 was the lowest and was located in the third category, indicating that its ability to absorb and accumulate nutrient elements in the fruit was relatively weak. We can also see from the actual hanging fruits in the orchard that the fruits of O.C. and GR1, planted in the rocky desert mountainous area, were large and uniform. In contrast, the fruits of H2 appeared to be of uneven sizes, which may also be a result of the unbalanced nutrient absorption of the fruits. According to the principle of the maximum coefficient of determination R2 and $p < 0.05$, the regression equation D = $-1.1794 + 0.0351P + 0.0738K + 0.0221Fe + 0.0019Mn + 0.0877Cu$ was obtained, with R2 = 0.997 and $p = 0.0003$. According to this equation, P and K are key to the mineral nutrition of the macadamia nut kernel, playing the role of indicators of major elements. At the same time, Fe, Mn, and Cu are very important indicators of trace elements.

With the development of the global macadamia nut industry, the nutritional quality of the fruit and the comprehensive utilization of the by-products are becoming increasingly important. This study shows that macadamia nuts are rich in N, P, K, Ca, and Mg, as well as microelements, but there are significant differences among different cultivars. Therefore, selecting and breeding some cultivars with strong adaptability, nutrient absorption, and accumulation capacity for the unique karst rocky desertification mountainous areas in Guizhou is an important direction for developing the macadamia nut industry in rocky desertification regions. This study found that the mineral elements in the pericarp of macadamia nuts grown in rocky desertification are rich in mineral contents. Therefore, it is of great research significance to fully utilize pericarp by-products, especially in China, the country with the largest planting area of macadamia nuts. The total weight of the pericarp and hulls after the bumper production can reach 600,000 tons, so applying the pericarp as a raw material for organic fertilizer is an important prospect.

## 5. Conclusions

Macadamia nuts grown in rocky desert mountains are rich in crude fat, protein, and soluble sugar, and the crude fat content is relatively stable among different varieties. Still, the soluble sugar content varies greatly. At the same time, the by-products of macadamia nuts' fruit peel are rich in mineral elements, which have greater development and utilization value. Our comprehensive evaluation of mineral elements in nine macadamia nut kernels showed that the O.C, 788, and GR1 kernels had the highest comprehensive mineral element contents, which indicated that they had a strong ability to absorb and accumulate mineral elements in rocky desertified mountains. At the same time, combined with the yields of different cultivars, we believe that varieties such as O.C and GR1 are excellent cultivars that can be popularized for planting in rocky, desertified mountains.

**Author Contributions:** All authors contributed to the study's conception. Material preparation was performed by G.G. and F.H. Z.K. analyzed the data, constructed the figures, and wrote the draft. W.W., H.Z. and X.T. conceived and designed the experiment, provided financial support, and reviewed the manuscript. All authors have read and agreed to the published version of the manuscript.

**Funding:** This research was financially supported by The Guizhou Forestry Scientific Research Project (Qian Lin Ke He [2020]03), Forestry Science and Technology Innovation Platform Operation Subsidy Funds (2020132540), Research on the purification and enrichment technology of ω-7 fatty acids in macadamia nut oil (Youth Fund of Guizhou Academy of Agricultural Sciences [2022] No. 34),

Subsidies for the Cultivation of Forest Tree Seeds and Seedlings (2024-GZZM-KBM-01), Research on Key Technologies of Germplasm Innovation and High Efficiency of Characteristic Crops in Guizhou Hot Area (Guizhou Agricultural Germplasm Resources No.8, 2023) and The National Key Research and Development Program of China, research and demonstration of key technologies for cultivating new tropical woody oil crop varieties and efficient matching (2023YFD2200705).

**Data Availability Statement:** All data are contained within this article.

**Acknowledgments:** All authors have acknowledged the content of the article and agreed to this publication.

**Conflicts of Interest:** The authors declare that they have no known competing financial interests or personal relationships that could have appeared to influence the work reported in this paper.

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
