# Peer review of "An Analysis of the Main Nutrient Components of the Fruits of Different Macadamia (Macadamia integrifolia) Cultivars in Rocky Desertification Areas and a Comprehensive Evaluation of the Mineral Element Contents"

_horticulturae, doi:10.3390/horticulturae10050468_

Round 1

Reviewer 1 Report

Comments and Suggestions for Authors

Comments on the Quality of English Language

Author Response

Thank you for your comment. We have revised the title and introduction, which are highlighted in red font throughout the article. The reasons for the formation of rocky desertification and the significance of conducting this study were added (In lines 45-78, 75-78). In the meantime, we have revised some parts of the discussion. Since, the main focus of this study is on the mineral content of the fruit, we really can't address some of the comments you made, but we will improve them in the next study.  

Finally, thank you for your arduous work and instructive advice. Special thanks to you for your good comments.

Reviewer 2 Report

Comments and Suggestions for Authors

The article entitled "Comprehensive evaluation of fruit mineral nutrition of different macadamia cultivars in rocky desertification areas" presents its merit, however it needs to be systematically restructured, so that the introduction brings information to better contextualize the study, in addition it needs to add quality assessments of the fruit once it was contemplated, and also in order to reach the reported conclusion, it is necessary to carry out analyzes of the concentration of mineral nutrients in the leaves.

The other allocations are described in the pdf.

Author Response

1. It needs to be systematically restructured, so that the introduction brings information to better contextualize the study, in addition it needs to add quality assessments of the fruit once it was contemplated.

Response: Thank you for your comment. We have revised this section in accordance with the suggestions provided by the reviewer. We revised the introductory section to highlight the purpose and significance of conducting this study (In lines 47-70). At the same time, we added the content of major nutrients in macadamia nuts kernels, specifically crude fat, crude protein and total soluble sugar content (In lines 135-156).

2. It is necessary to carry out analyzes of the concentration of mineral nutrients in the leaves.

Response: Thank you for your comment. Mineral nutrition of leaves is very important and this is where we need to focus our research next, but in this study we mainly focused on the nutritional status of fruits. In the later work we will add and improve according to the reviewers' comments.

Finally, thank you for your arduous work and instructive advice. Special thanks to you for your good comments.

Reviewer 3 Report

Comments and Suggestions for Authors

This paper aim is to evaluate the differences in fruit mineral element contents of different macadamia cultivars and to screen out good cultivars suitable for promotion in rocky desert mountains. Nine macadamia nut cultivars were selected as test materials in rocky desert mountain orchards. It is interesting paper and very good presents the results of work.

Macadamia is grown for its tasty nuts, the kernel of which consists of 77% of fats, including monounsaturated acids, protein and vitamins, especially B group and vitamin A and E. They contain minerals important for the proper functioning of the body: calcium, phosphorus, potassium, iron.

The work has many scientific values, i.e. good prepared methodical of work, perfect presentation of the results and statistical analysis. In my opinion, the results are not comprehensive. The Authors present contents of N, P, K, Ca, Mg, Fe, Mn, Cu, Zn, and B in the peel and kernel were determined. The work should also be supplemented with other ingredients of these nuts, for example the content of monounsaturated acids, protein or some vitamins. The value of work would then increase greatly.

Author Response

The work should also be supplemented with other ingredients of these nuts, for example the content of monounsaturated acids, protein or some vitamins. The value of work would then increase greatly.

Response:Thank you for your comment. We acknowledge the limitations of our study, we supplemented data on crude fat, crude protein, and total soluble sugar content of different macadamia nut varieties.

Finally, thank you for your arduous work and instructive advice. Special thanks to you for your good comments.

Reviewer 4 Report

Comments and Suggestions for Authors

Brief summary

Interesting manuscript that addresses the study of principal components as a technique to study the selection by nutrient efficiency of nine macadamia cultivars and the genetic adaptability to conditions in rocky areas of desertification.

General concept comments

It is mandatory for authors and a cover letter should be included with each manuscript submission explaining why the content of the paper is significant, placing the findings in the context of existing work, and why the manuscript fits the aims and scope of the journal.

It is also mandatory to adapt the manuscript to the instructions for the authors of Horticulturae. This is especially recommended for bibliographic references, author affiliation and manuscript sections.

https://www.mdpi.com/journal/horticulturae/instructions

Despite presenting interesting information about the elemental content of the peel and kernel of the fruits, the work is far from being a “comprehensive assessment of mineral nutrition”, so the title can be pretentious and generally confusing to readers.

The introduction adequately presents the situation of the introduction of macadamia cultivation in China, but lacks an explicit hypothesis and does not adequately present the objectives of the study.

Material and methods adequately present the data processing to perform the analysis of principal components, but the analytical procedures in a concise and non-standardized manner.

Results presents tables on the elemental content of the macadamia nut, both the core and the shell. However, it would be advisable to express the results in % of dry weight for macronutrients and in ppm for micronutrients. Correlation analyses, principal components, affiliation analysis, function values, and cluster analysis can be considered suitable tools for establishing an adaptability ranking. Section “3.5.5 Characterization of the elemental content in leaves?” does not have foliar content values (which have not been determined). Harvest data are not presented for the different cultivars. This deficiency may negate the conclusion regarding the selection of cultivars “such as O.C, GR1 and A16 are excellent cultivars that can be popularized for planting in rocky desert mountains”.

Discussion should emphasize the possibility of using this methodology to assess the nutritional quality of the fruit and the integral utilization of the by-products and the selection of some cultivars with strong adaptability, nutrient absorption and accumulation capacity for the unique mountainous areas of karst rocky desertification in the development of the industry

Comments on the Quality of English Language

A revision of English is necessary, especially in the definition of the sections of the manuscript in a more concrete way.

Author Response

1. Despite presenting interesting information about the elemental content of the peel and kernel of the fruits, the work is far from being a “comprehensive assessment of mineral nutrition”, so the title can be pretentious and generally confusing to readers.

Response: Thank you for you comment. We have changed the title of the paper to "Analysis of the main nutrient components of the fruits of different macadamia cultivars in rocky desertification areas and comprehensive evaluation of the mineral element contents".

2. The introduction adequately presents the situation of the introduction of macadamia cultivation in China, but lacks an explicit hypothesis and does not adequately present the objectives of the study.

Response: Thank you for you comment. We have revised this section in accordance with the suggestions provided by the reviewer. We revised the introductory section to highlight the purpose and significance of conducting this study (In lines 47-70).

3. Section “5.5 Characterization of the elemental content in leaves?” does not have foliar content values (which have not been determined). Harvest data are not presented for the different cultivars.

Response: Thank you for you comment. Here we would like to discuss the content of mineral elements of kernels in different classifications. We are not discussing the mineral content of leaves, which is due to an error in our work, which we have revised.

4. Discussion should emphasize the possibility of using this methodology to assess the nutritional quality of the fruit and the integral utilization of the by-products and the selection of some cultivars with strong adaptability, nutrient absorption and accumulation capacity for the unique mountainous areas of karst rocky desertification in the development of the industry.

Response: Thank you for you comment. In response to the question of by-product utilization, we re-explained it in our discussion (In lines 351-357).

5. A revision of English is necessary, especially in the definition of the sections of the manuscript in a more concrete way.

Response: Thank you for you comment. We touched up the full text through the MDPI platform in order to achieve the level of linguistic expression required for journal publication.

Finally, thank you for your arduous work and instructive advice. Special thanks to you for your good comments.

Round 2

Reviewer 2 Report

Comments and Suggestions for Authors

After this systemic restructuring of work. I agree with the publication.

Author Response

Thank you very much for your guidance on this article

Reviewer 3 Report

Comments and Suggestions for Authors

The Author fast and smart present the main nutrient composition of kernels of different macadamia nut cultivars.   In this study, The Authors found that all the macadamia nut cultivars with a crude fat content above 75% reached the expected target, which meets the basic requirements for the quality of macadamia nut commercial fruits, indicating that this region is suitable for macadamia nut cultivation. The conclusion, from the point of view of the main nutritional components of macadamia nuts, it is feasible to plant macadamia nuts in rocky desertification areas.

But the results of this analysis should be taken in to the PCA and cluster analysis. 

Author Response

The results of this analysis should be taken in to the PCA and cluster analysis.

Response: Thank you for you comment. Our main focus in this article was on the mineral element content of fruits in macadamia orchards in rocky desertified mountains, therefore, indicators such as soluble sugars, soluble proteins, and crude fats were not included inside the principal component analysis..

Finally, thank you for your arduous work and instructive advice. Special thanks to you for your good comments.

Reviewer 4 Report

Comments and Suggestions for Authors

General concept comments

Many paragraphs in the manuscript are excessively long. The sentences in each paragraph are also excessively long (which makes them difficult to read and understand). It would be very convenient to submit the manuscript to the option: Submit a new manuscript for English editing at https://www.mdpi.com/authors/english.

The titles of tables and figures should fully describe their contents, even if they are separated from the manuscript text. The foot of tables and figures should not be ambiguous or make non-specific references to the text ("same as below")

Some paragraphs of results appear duplicates in the discussion. To avoid this duplication, it would be advisable to present a single section in the manuscript with results and discussion.

To facilitate the location of the references used, it is advisable to provide the DOI of the publications that have it. It would be advisable to use a bibliographic reference manager for adequate management of the bibliography and use the bibliographic style appropriate for Horticulturae.

Specific comments

Specific comments can be found in the attached document

Comments on the Quality of English Language

Many paragraphs in the manuscript are excessively long. The sentences in each paragraph are also excessively long (which makes them difficult to read and understand). It would be very convenient to submit the manuscript to the option: Submit a new manuscript for English editing at https://www.mdpi.com/authors/english.

Author Response

Questions about English editing.

Response: Thank you for you comment. We touched up the full text through the MDPI platform in order to achieve the level of linguistic expression required for journal publication. The article's English editor's ID number is 79360.

To facilitate the location of the references used, it is advisable to provide the DOI of the publications that have it.

Response: Thank you for you comment. We have included links to download the literature at the end of the references, as well as DOIs for the Chinese references; however, we apologize that some of the Chinese references do not have DOI numbers.